# Serum Lipocalin-2 Levels as a Biomarker in Pre- and Post-Pubertal Klinefelter Syndrome Patients: A Pilot Study

**DOI:** 10.3390/ijms25042214

**Published:** 2024-02-12

**Authors:** Roberto Paparella, Giampiero Ferraguti, Marco Fiore, Michela Menghi, Ginevra Micangeli, Francesca Tarani, Aurora Ligotino, Marisa Patrizia Messina, Mauro Ceccanti, Antonio Minni, Christian Barbato, Marco Lucarelli, Luigi Tarani, Carla Petrella

**Affiliations:** 1Department of Maternal Infantile and Urological Sciences, Sapienza University of Rome, 00161 Rome, Italy; roberto.paparella@uniroma1.it (R.P.); michela.menghi@uniroma1.it (M.M.); ginevra.micangeli@uniroma1.it (G.M.); francesca.tarani@uniroma1.it (F.T.); marisapatrizia.messina@uniroma1.it (M.P.M.); luigi.tarani@uniroma1.it (L.T.); 2Department of Experimental Medicine, Sapienza University of Rome, 00161 Rome, Italy; giampiero.ferraguti@uniroma1.it (G.F.); marco.lucarelli@uniroma1.it (M.L.); 3Institute of Biochemistry and Cell Biology (IBBC-CNR), Policlinico Umberto I, 00161 Roma, Italy; marco.fiore@cnr.it (M.F.); christian.barbato@cnr.it (C.B.); 4SITAC, Società Italiana per il Trattamento Dell’alcolismo e le sue Complicanze, 00185 Rome, Italy; mauro.ceccanti@uniroma1.it; 5Department of Sensory Organs, Sapienza University of Rome, 00161 Roma, Italy; antonio.minni@uniroma1.it; 6Division of Otolaryngology-Head and Neck Surgery, San Camillo de Lellis Hospital, ASL Rieti-Sapienza University, 02100 Rieti, Italy; 7Pasteur Institute Cenci Bolognetti Foundation, Sapienza University of Rome, 00161 Roma, Italy

**Keywords:** lipocalin-2, Klinefelter syndrome, infertility, metabolic disorders, LH, inhibin B, HDL

## Abstract

Klinefelter syndrome (KS) is a male genetic disease caused by the presence of an extra X chromosome, causing endocrine disorders mainly responsible for a high rate of infertility and metabolic disorders in adulthood. Scientific research is interested in identifying new biomarkers that can be predictive or prognostic of alterations strictly connected to KS. Lipocalin-2 (LCN-2, also known as NGAL) is a small protein initially identified within neutrophils as a protein related to innate immunity. Serum LCN-2 estimation seems to be a useful tool in predicting the metabolic complications caused by several pathological conditions. However, little is known about its potential role in infertility conditions. The present pilot study aims to investigate the presence of LCN-2 in the serum of a group of pre-pubertal and post-pubertal children affected by KS, compared to healthy controls. We demonstrated for the first time the presence of elevated levels of LCN-2 in the serum of KS patients, compared to controls. This increase was accompanied, in pre-pubertal KS patients, by the loss of correlation with LH and HDL, which instead was present in the healthy individuals. Moreover, in all KS individuals, a positive correlation between LCN-2 and inhibin B serum concentration was found. Despite the limited size of the sample analyzed, our preliminary data encourage further studies to confirm the findings and to extend the study to KS adult patients, to verify the predictive/prognostic value of LCN-2 as new biomarker for metabolic diseases and infertility associated with the pathology.

## 1. Introduction

Klinefelter syndrome (KS) is a genetic disease caused by the presence of an extra X chromosome in male individuals [1,2]. The most frequent karyotype is 47,XXY, even if mosaicism could be present. This chromosomal set causes endocrine disorders that affect testicular function, often manifesting in adulthood as hypogonadism and high rate of infertility due to poor sperm production or quality [2]. KS-affected patients display abnormal or delayed development of typically male sexual characteristics [3]. The testicles are generally small, and gynecomastia also frequently arises. In contrast, testicular histology obtained from pre-pubertal KS patients appears to be nearly normal, with primary germ cells being present [4], confirming that, in many cases, a KS diagnosis can be delayed until adulthood, being found only during investigations for infertility [5].

A certain percentage of KS adult subjects suffer from metabolic syndrome, a clinical condition characterized by the simultaneous presence of lipid and glucose metabolism dysfunctions with an associated increase in cardiovascular risk [6,7,8]. Already during pre-pubertal age, KS individuals may display higher adiposity compared to their healthy peers, despite similar, physiologically low, serum testosterone levels [2,9].

Lipocalin-2 (LCN-2, also known as NGAL) is a small protein (about 25 Dalton) which belongs to the lipocalin family. LCN-2 was initially identified within neutrophils as a protein related to innate immunity. This protein is synthesized by neutrophils following the activation of toll-like receptors and its role is to bind iron, removing it from the bacteria and inhibiting their growth [10]. LCN-2 is constitutively expressed in several tissues including the kidney, lung, bone marrow, liver, adipose tissue, macrophages, thymus, breast duct, prostate, small intestine and trachea [11,12]. Interestingly, LCN-2 expression may be induced in tissue where it is normally absent (brain, heart, skeletal muscle, spleen, testes, ovary and colon) under different pathological conditions. Several studies support the role of LCN-2 serum estimation as a useful tool in predicting the metabolic complications caused by obesity [13,14], in conditions arising from intestinal disorders [15,16] and in patients with cardiovascular complications [17,18]. However, little is known about its potential role in infertility conditions [11].

Scientific research is interested in identifying new biomarkers that can be predictive or prognostic of alterations strictly connected to KS (infertility, metabolic dysfunctions) [19,20,21,22].

The present pilot study was aimed at investigating the presence of LCN-2 in the serum of a group of pre-pubertal and post-pubertal children affected by KS, compared to age-matched healthy controls. We also correlated these results with the main biochemical parameters associated with sexual development (testosterone, FSH, LH, inhibin B), lipid (triglycerides, total cholesterol, HDL and LDL) and glucose (glycemia, insulinemia, HOMAi) metabolism. We predict that we will demonstrate, for the first time, the presence of LCN-2 in the serum of KS patients, whose levels are significantly elevated when compared to age-matched healthy controls. This increase is accompanied, in pre-pubertal KS patients, by the loss of correlation with LH and HDL, which instead, is present in healthy subjects. Moreover, in all KS individuals, a positive correlation between LCN-2 and inhibin B serum concentration was found.

## 2. Results

### 2.1. Lipocalin-2 Serum Levels

As shown in Figure 1, KS individuals showed higher levels of serum LCN-2 in comparison to healthy controls [F (1, 49) = 11.52; *p* = 0.0014] (small panel). Post hoc analysis revealed that this difference was mainly due to the pre-pubertal children, as shown in the graph (large panel) (*p* = 0.019).

### 2.2. Metabolic Assessments 

Serum from healthy controls and KS patients were analyzed to compare the clinical parameters concerning sex hormone and metabolic (glucose and lipidic metabolism) profiles.

#### 2.2.1. Sex Hormone Profile

As shown in Figure 2, testosterone levels were higher in post-pubertal children, both in controls and KS patients (eugonadal) [F (1, 48) = 100.3; *p* < 0.0001]. Concerning FSH levels, two-way ANOVA revealed a “KS” effect [F (1, 48) = 10.59; *p* = 0.0021] due to an increase in FSH levels in KS patients compared to healthy controls [age x disease, F (1, 48) = 8.298; *p* = 0.0059]. Moreover, the effect of age was revealed in post-pubertal individuals [F (1, 48) = 13.18; *p* = 0.0007] because of the expected increase in FSH values in post-pubertal KS children with respect to age-matched healthy controls.

Finally, elevated LH serum levels in post-pubertal compared to pre-pubertal groups [F (1, 48) = 46.76; *p* < 0.0001], particularly in KS children (*p* < 0.001) were disclosed in the statistical analysis.

In pre-pubertal KS children, serum inhibin B, an important marker for testicular functioning, was in the physiological range, even though a subgroup of patients exhibited lower than normal values. Post-pubertal KS patients showed reduced levels of inhibin B, indicating testicular impairment, which was not significant and probably due to values still being within the normal range.

#### 2.2.2. Lipid Profile

In Figure 3, the lipid profile of the recruited individuals is reported. Total cholesterol levels were comparable between all groups, with no changes due to age and/or disease. though if they were within the normal range, HDL serum levels were increased in KS children when compared to controls [F (1, 48) = 26.08; *p* < 0.0001]. Moreover, multiple comparisons revealed significant differences between pre- and post-pubertal groups (see post hoc analysis in the graph). Concerning LDL values, a significant difference was observed when healthy controls and KS patients were compared [F (1, 48) = 5.618; *p* = 0.0218], without any age-related effects. Finally, a disease-dependent effect was observed on triglyceride serum levels in KS children when compared to healthy controls [F (1, 47) = 15.95; *p* = 0.0002]. A strong significant reduction was detected in post-pubertal KS children with respect to healthy controls (*p* = 0.016). Interestingly, the mean value of triglycerides in post-pubertal KS patients was under the lower value of the normal range. In pre-pubertal children, we observed a tendency, though not a significant one, towards lower serum values in KS patients (*p* = 0.06).

#### 2.2.3. Glucose Profile 

The glucose profiles are shown in Figure 4. KS patients showed higher glucose levels in comparison to healthy controls, even though they are within the range value of normality [F (1,44) = 6.788; *p* = 0.0125]. An interesting, significant, interaction between controls and KS patients was observed due to a reduction (in controls) and an elevation (in KS) in glucose levels, which was age-dependent [F (1, 44) = 5.792; *p* = 0.0204]. Concerning the insulin values, an effect of age was observed, with pre-pubertal values being lower than post-pubertal ones [F (1, 43) = 8.821; *p* = 0.0049].

Finally, HOMAi was significantly higher in post-puberal vs. pre-pubertal KS individuals [F (1, 44) = 11.90; *p* = 0.0014], showing a predisposition to insulin resistance in KS patients.

### 2.3. Spearman Correlation LCN-2 vs. Clinical/Metabolic Parameters 

In Table 1, the main Spearman correlation analyses between LCN-2 and clinical/metabolic parameters were reported.

In healthy pre-pubertal controls, significant negative correlations between LCN-2 and HDL and LH were observed (*p* = 0.003 and *p* = 0.025, respectively): for high values of LCN-2, low values of HDL and LH were found. On the contrary, no correlations were disclosed in pre-pubertal KS patients between LCN-2 and the analyzed parameters (*p* = 0.423 and *p* = 0.959, respectively). Moreover, a positive correlation existed between LCN-2 and inhibin B in all KS individuals (pre- and post-pubertal) (*p* = 0.045), which was probably mainly due to the prepubertal group (*p* = 0.059).

## 3. Discussion

In this pilot study, we demonstrated, for the first time, the presence of LCN-2 in the serum of pre-pubertal and post-pubertal individuals affected by KS. We found increased circulating levels in all KS individuals, compared with healthy controls, which were more evident in pre-pubertal children. This clinical study also demonstrated that pre-pubertal KS patients have increased LCN-2 levels in the serum compared to age-matched controls, which preceded the well-known detectable changes in serum testosterone, FSH, LH and inhibin B levels with this growth.

The ubiquitous presence of LCN-2 in different cellular districts of the human body makes it an interesting candidate as a marker of various pathological conditions, not only for metabolic syndrome. The latter refers to the presence of a cluster of risk factors for the development of cardiovascular pathologies, concerning alterations in glucose and/or lipid metabolism (diabetes, obesity, hypertension). Klinefelter syndrome is characterized by infertility due to the failure of the sperm genesis process accompanied by alterations to the hypothalamic–pituitary–gonadal axis. Klinefelter patients are also considered at risk for developing metabolic syndrome, as the altered chromosomal structure can predispose patients, especially adults, to changes in the lipid, glucose and sex-related hormone profiles. The elevation in circulating levels of LCN-2 in KS patients constitutes a starting point for investigating the reason for this alteration, starting from the potential relationship with the main biochemical and clinical parameters specific to the group of patients enrolled in the study.

KS patients enrolled in the study were characterized by specific sex hormone, lipidic and glucose metabolic profiles, that we correlated with the LCN-2 levels, to disclose the putative link between the protein and biological functions.

Firstly, KS patients showed age-dependent comparable testosterone and LH levels with respect to healthy controls. Moreover, increased levels of FSH in post-pubertal KS patients, as expected in hypergonadotropic hypogonadism, were found. Finally, post-pubertal KS patients had low levels of inhibin B, which inversely correlated with FSH concentrations, well reflecting spermatogenetic damage. In our pilot study, LCN-2 levels were positively correlated with inhibin B, mainly in pre-pubertal KS patients. Moreover, LCN-2 levels were inversely correlated to LH levels only in healthy pre-pubertal children, but not in KS age-matched patients, supporting the involvement of LCN-2 system in the homeostasis of human reproduction. To date, several studies, in animal models and in vitro experiments, have investigated the role of LCN-2 in the male reproductive system and fertility. The results of these studies do not offer a single interpretation of the potential role of the LCN-2 system, leaving further hypotheses and investigations open. Two interesting preclinical reports have focused on the identification of the genes that participate in the dialogue existing between germ cells and Sertoli cells, which are crucial for the correct process of spermatogenesis [23,24]. Authors have demonstrated that, in sterile strain mice, in the early stages of testicular development, germ cells and Sertoli cells communicate through molecular signals, modulating the expression of a group of genes, including *lcn-2*. They determined that spermatogonia cell-derived factors positively regulate the expression of the *lcn-2* gene in Sertoli cells, and that the NF-kB pathway was involved in the transcriptional activation. However, the exact role of LCN-2 in the male reproductive system is not clear. Intriguingly, an in vitro assay demonstrated that mouse LCN-2 enhanced sperm motility, increasing intracellular cAMP concentration [25] and participating in regulating flagellar motility. The LCN-2 protein was also found to be expressed in the epididymis of 14-day-old mice up to the complete epididymal differentiation. Mouse epididymal spermatozoa exhibited the ability to bind with exogenous LCN-2, a part of which is further incorporated into the cytoplasmatic compartment, mediating, via protein internalization, the delivery of ferric ions, which are essential for spermatozoa functionality [26,27,28]. More recently, the LCN-2 protein has been identified in mouse Leydig cells, with an age-related change in its levels during life. In particular, increased expression was observed up to 8 months of age, with a subsequent decrease in old mice, reproducing the same age-dependent trend seen in testosterone levels [29]. Interestingly, the authors also observed augmented LCN-2 mRNA and protein levels in the testes of two different infertile mouse models (induction of cryptorchidism and busulfan-led induction of apoptosis in germ cells). LCN-2 increase correlated with the reduction in the number of germ cells, probably associated with the ability of LCN-2 in regulating apoptotic processes [29,30].

Our pilot study is the first to display a modification in serum LCN-2 in association with reproduction/infertility in male individuals. Even if we cannot establish the exact role, the inverse correlation between LCN-2 and LH, in pre-pubertal healthy controls, supports the hypothesis of an early physiological role of LCN-2 in male sexual development, in which low levels of LCN-2 and high levels of LH favor the spermatogenesis process. In view of the loss of this correlation in KS children, the augmented LCN-2 serum levels observed in KS patients could be the effect of the lack of a homeostasis mechanism.

In KS patients we observed an inverse correlation between LCN-2 and inhibin B serum levels, mainly in pre-pubertal patients. Inhibin B is a heterodimeric glycoprotein hormone secreted by Sertoli cells in the testes which inhibits the production of the FSH hormone which acts indirectly on the development of male gametes. The dosage of inhibin B in men can be used as a marker of spermatogenic function and male fertility. In particular, very low levels of inhibin B indicate insufficient or no production of spermatozoa and have proven to be a good predictive signal of success/failure of the surgical recovery of sperm for the subsequent performance of in vitro fertilization [31]. In KS patients, the positive correlation between LCN-2 and inhibin B could reflect the potential value of this protein as a marker indicating the functional impairment of Sertoli cells in this pathology. Remarkably, the evidence of a more significant relationship between LCN-2 and inhibin B confined to pre-pubertal age (when FSH values are comparable to those of healthy individuals) makes LCN-2 a potential predictor of functional defect of the reproductive system.

The analysis of the lipidic profile in the enrolled patients and controls displayed an effect of the pathology on circulating LDL, HDL and triglycerides, when compared to healthy controls. Surprisingly, in our cohort of patients, reduced LDL and high HDL levels vs. those of healthy controls were observed, describing a favorable metabolic picture. However, the very low circulating triglycerides observed in KS patients might represent an early alteration in the machine of lipid turnover. Conversely, neither a disease- nor age-dependent effect was observed on total cholesterol serum levels, which were similar in all groups. Concerning glucose profile, we detected raised glycemia values (even if within the physiological range) in post-pubertal KS patients, which is associated with an increase in HOMAi, both predisposing to insulin resistance.

Several studies, in pre-clinical models and in humans, have revealed LCN-2 as biomarker in obesity, insulin resistance and hyperglycemia [14]. LCN-2 was associated with adverse lipid profiles, but independent of age, sex and adiposity, suggesting that this protein might be an autonomous risk factor for hyperglycemia and insulin resistance in humans [14]. There have been some controversial results originating from animal studies. Lcn-2KO mice predispose to the progression of spontaneous age-related adiposity in mice, accelerating weight gain and visceral fat deposition with age, when compared to wild-type mice [32] On the other hand, in diabetic/obese mice and their age- and sex-matched lean littermates, LCN-2 mRNA levels in adipose tissue and liver and its circulating concentrations were significantly increased with respect to wild-type mice [14]. In our pilot study, we found an inverse correlation between LCN-2 and HDL in pre-pubertal healthy controls, lost in age-matched KS children, that showed how, early in life, the salutary HDL value corresponded to low LCN-2 levels.

The “increased cardiovascular risk” in KS patients affected by metabolic syndrome is observed in a variable percentage of patients as demonstrated in several studies, mainly in adults [33,34,35] and few others in children [36,37]. The increased probability often depends on the simultaneous presence of different risk factors. In particular, hypogonadism and obesity are strictly linked to the onset of cardiovascular complications associated with metabolic syndrome [33,38]. In our cohort of patients, testosterone levels, as well as BMI, are in the normal range values and in line with a favorable overall metabolic picture. Nonetheless, we cannot exclude the possibility that the hormonal picture (lipid or sex-related) may change in adulthood and leave room for modifications in the risk of developing metabolic syndrome.

As mentioned before, LCN-2 is a neutrophil gelatinase-associated lipocalin that also plays a role in innate immunity by limiting bacterial growth as a result of sequestering iron-containing siderophores [39]. The iron-bound form (holo-24p3) is, in fact, internalized following binding to the SLC22A17 receptor (24p3R), leading to a decrease in extracellular iron concentration [39] that disfavors bacterial growth. Autoimmune diseases, dysfunctions of the immune system that recognizes the “self” as “non-self”, can be triggered by various factors, including antigens of bacterial origin which, due to the similarity with elements of the organism, mistakenly trigger a defensive response against one’s own organism. Autoimmune diseases are more common among women [40]. Interestingly, KS individuals have a higher risk of developing auto-immune disorders, compared to the general population. The mechanism underlying this phenomenon is unclear. Possible cofactors that come into play are gonadal hormonal imbalance (hypogonadism) or effects directly related to the X chromosome asset. A recent study, investigating the presence of auto-antibodies, showed that, in 47,XXY KS patients, ANAs (anti-nucleus antibodies) were observed significantly more frequently than in the controls [41]. Given these premises, it could be very interesting to investigate whether LCN-2 levels in KS patients can be correlated with the presence or be predictive of the risk of onset of autoimmune disorders.

The main limitation of the present study is the limited size of the sample analyzed, along with the lack of data addressing the mechanisms underlying the increase in circulating concentrations of LCN-2 in KS patients. Furthermore, we cannot state whether the increase in LCN-2 is a causal factor or simply an independent factor in the pathogenesis of the metabolic and infertility abnormalities associated with the syndrome.

Nonetheless, our preliminary data encourage further studies to support the correct direction of the observed associations. It would also be interesting to extend the study to an adult population of KS patients, typically characterized by alterations in their hormonal (hypogonadism) and metabolic pathways that are much more evident than in young individuals, in order to verify the predictive/prognostic value of this new biomarker for metabolic diseases and infertility associated with the pathology.

## 4. Materials and Methods

### 4.1. Patients Enrollment

Thirty-one KS subjects with classic 47,XXY karyotype, were enrolled at the Department of Maternal Infantile and Urological Sciences, Sapienza University Hospital “Policlinico Umberto I” of Rome, Italy. The 47,XXY karyotype was proven by an analysis of 15 to 30 metaphases with G-banding and/or an analysis of 50 ± 100 interphases with FISH analysis in peripheral blood preparations and/or fibroblasts [42]. For the study, KS individuals were divided into two subgroups: pre-pubertal (n = 16, mean age 8.3 ± 0.57) and post-pubertal (n = 15, mean age 16.3 ± 0.70) KS children. A total of 23 healthy age-matched controls [pre-pubertal (n = 9, mean age 8.7 ± 0.8) and post-pubertal (n = 14, mean age 18.3 ± 1.8)] were recruited at the Sapienza University Hospital pediatric clinic. Children were selected as they attended the hospital for the investigation of the presumed pathologies that resulted in not being present at all, defining thus the children as “healthy” [20]. Pre-pubertal KS vs. pre-pubertal healthy prepubertal individuals (18.8 ± 1.4 *vs* 15.4 ± 2.9), and post-pubertal KS vs. post-pubertal controls (22.2 ± 1.4 vs 22.2 ± 0.9), had comparable body mass indexes.

The exclusion criteria for all the individuals enrolled were: (1) the presence of genetic mosaicism (for KS subjects), (2) the use of drugs that could influence the serum levels of LCN-2 (antidepressant, immunosuppressant or immunomodulatory drugs), (3) inflammatory, endocrine, cardiovascular and autoimmune diseases not strictly related to the syndrome that could influence the serum levels of LCN-2, (4) other systemic disorders that could influence the serum levels of LCN-2.

The study was approved by the Sapienza University Hospital ethical committee (Ref. 5825); all the study procedures were under the Helsinki Declaration of 1975, as revised in 1983, for human experimentation.

Anthropometric and body composition data (age, height, weight, body mass index/BMI) were collected for all participants. Peripheral blood samples of 5 mL were taken from each participant, collected in BD Vacutainer™ Serum Separation Tubes and centrifuged at 3000× *g* rpm for 15 min to separate serum from blood cells. The serum was stored at −20 °C till the day of the analysis.

Metabolic parameters were evaluated in the laboratory of the Experimental Medicine Department, Section of Medical Pathophysiology, Food Science and Endocrinology of the Policlinico Umberto I. All the reference ranges reported referred to male individuals. Testosterone (reference range: 2.80–8 ng/mL), follicle-stimulating hormone (FSH) (reference range: 1.5–12.4 mUI/mL), luteinizing hormone (LH) (reference range: 1.7–8.6 mUI/mL) and inhibin B (reference range: 80–380 pg/mL) serum levels were evaluated by chemiluminescent microparticle immunoassay. Serum insulin (reference range: 2.6–24.9 μUI/mL), glucose (reference range: 70.3–100.9 mg/dL), triglycerides (reference range: 45.1–235.4 mg/dL), total cholesterol (reference range: 81.1–235.5 mg/dL) and high-density lipoprotein (HDL)-cholesterol (reference range: male 34.7–56.0 mg/dL) were measured using standard colorimetric and enzymatic methods on a Cobas C 501 analyzer. Low-density lipoprotein (LDL)-cholesterol levels (reference range: 54.8–120.1 mg/dL) were calculated using Friedwald equation [43]. The HOMA index (Homeostasis Model Assessment—HOMAi) was also calculated (reference range: 0.23–2.5) [44].

### 4.2. Serum Lipocalin-2 (LCN-2) Analysis

Human serum LCN-2 (Cat. No. DY1757) was measured using a sandwich enzyme-linked-immunosorbent assay (ELISA) kit (R&D Systems, Minneapolis, MN, USA), according to the protocols provided by the manufacturer. Briefly, the day before the test, a 96-well microplate was coated with the working concentration of capture antibody and incubated overnight at room temperature. The day of the test, after the washing process, the microplate was blocked with the reagent diluent for 1 h at room temperature, to saturate unspecific antigenic sites. 100 uL of standard curve and unknown samples were added in duplicate to each well and incubated for 2 h at room temperature. After repeating the washes, 100 uL of detection antibody were added in each well and incubated for 2 h at room temperature. Working solutions of Streptavidin-HRP were added to each well, avoiding placing the plate in direct light. The substrate solution developed a blue color in each well with an intensity directly proportional to the concentration of the protein of interest. An acidic solution was added to stop the reaction. Optical density of each well was determined using a microplate reader (NeoBiotech, Seoul, Republic of Korea) set to 450 nm. A standard curve has been created, by using a software able to generate a four-parameter logistic (4-PL) curve-fit. Final concentration of unknown samples has been extrapolated by using a data analysis tool (MyAssays; https://www.myassays.com/), considering the dilution factor (1:200). Data have been represented as ng/mL.

### 4.3. Statistical Analysis

Statistical significance was determined using the analysis of variance (ANOVA) test and *p* values less than 0.05 were considered as significant. Post hoc comparisons were performed by using the Tukey’s HSD test. The Spearman Correlation test was performed to investigate the association between serum LCN-2 levels with the clinical and metabolic parameters measured. Data were presented as mean ± standard error mean (SEM).

## Figures and Tables

**Figure 1 ijms-25-02214-f001:**
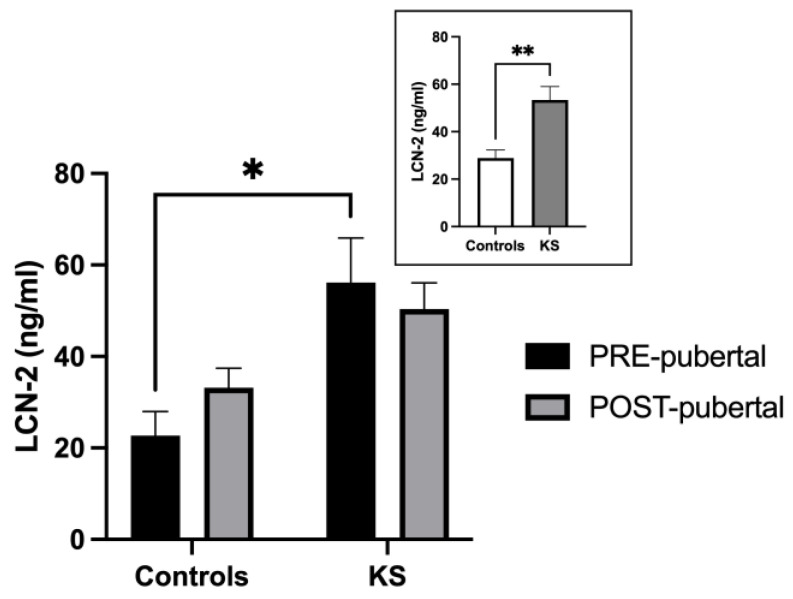
Histograms representing LCN-2 serum values (mean ± SEM) in all pre- and post-pubertal healthy controls and KS patients (small panel) and in pre- and post-pubertal subgroups (large panel). * *p* < 0.05; ** *p* < 0.01.

**Figure 2 ijms-25-02214-f002:**
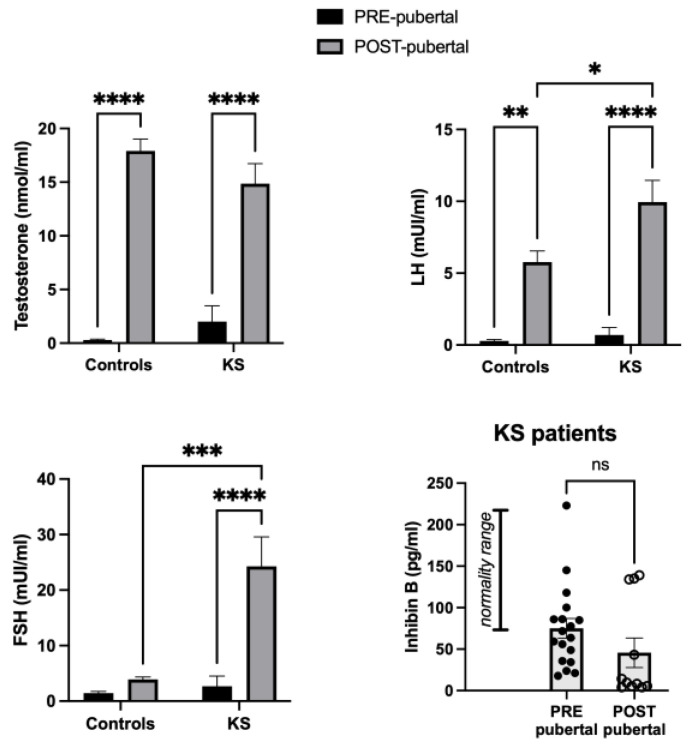
Histograms representing serum values (mean ± SEM) of sex hormone profile in pre- and post-pubertal healthy controls and KS patients. * *p* < 0.05; ** *p* < 0.01; *** *p* < 0.001; ***** p* < 0.0001; ns (not significant).

**Figure 3 ijms-25-02214-f003:**
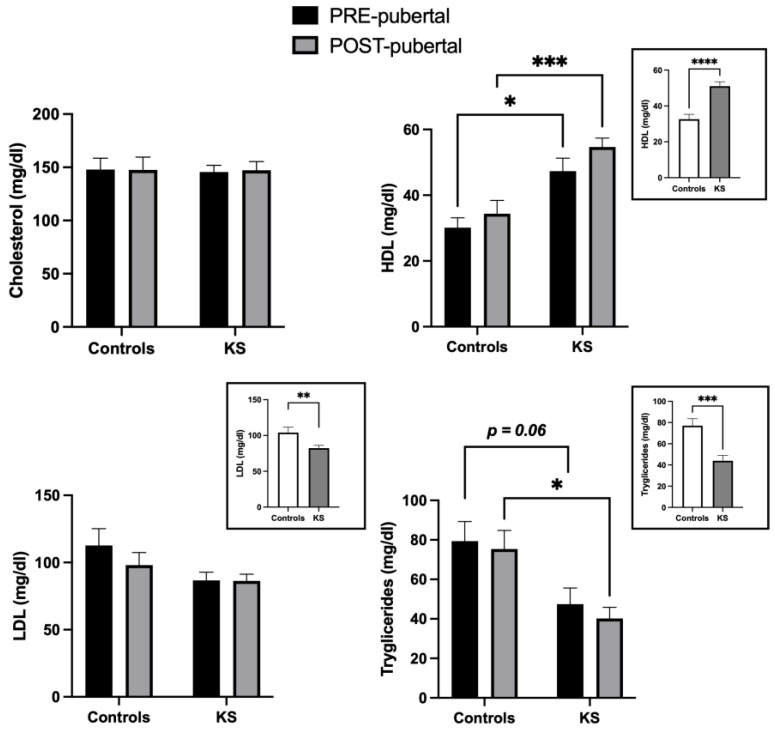
Histograms representing serum values (mean ± SEM) of lipid profile in all pre- and post-healthy controls and KS patients (small panel) and in pre- and post-pubertal subgroups (large panel). * *p* < 0.05; ** *p* < 0.01; *** *p* < 0.001; **** *p* < 0.0001.

**Figure 4 ijms-25-02214-f004:**
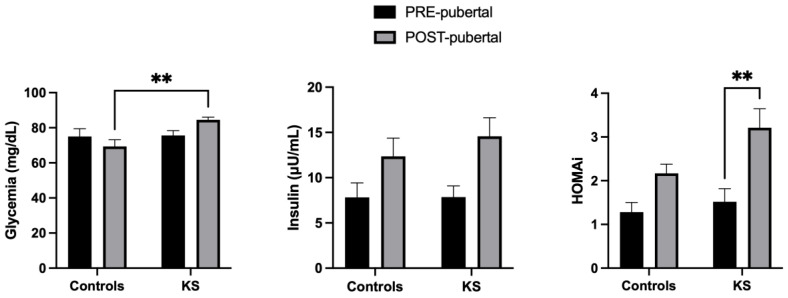
Histograms representing serum values (mean ± S.E.M) of glucose profile in pre- and post-pubertal healthy controls and KS patients. ** *p* < 0.01.

**Table 1 ijms-25-02214-t001:** Spearman correlation between LCN-2 (ng/mL) and the variable indicated. (*) indicates statistical significance.

Subjects		Variable	Correlation Coefficient (rho)	*p* Value
Healthy Controls	Pre-pubertal	HDL (mg/dL)	−0.854	0.003 *
KS individuals			0.223	0.423
Healthy Controls		LH (mUI/mL)	−0.73	0.025 *
KS individuals			0.014	0.959
KS individuals	All individuals	Inhibin B (pg/mL)	0.405	0.045 *
	Pre-pubertal		0.521	0.059
	Post-pubertal		0.26	0.441

No correlation was found between LCN-2 and all the other clinical and metabolic parameters examined (testosterone, FSH, total cholesterol, LDL, triglycerides, glycemia, insulinemia, HOMAi).

## Data Availability

Data are available on request due to ethical reasons.

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
