# Peer review of "Serum Lipocalin-2 Levels as a Biomarker in Pre- and Post-Pubertal Klinefelter Syndrome Patients: A Pilot Study"

_ijms, 2024, doi:10.3390/ijms25042214_

Round 1
Reviewer 1 Report
Comments and Suggestions for Authors
Thank you for the opportunity to read and review this interesting article titled "Serum Lipocalin-2 Levels as Biomarker in Pre- And Post-Pubertal Klinefelter Syndrome Patients: A Pilot Study" by Paparella et. al.
Topic of interest in the andrological field.
The article is well written with clearly exposed results, limitations, and future and potential future directions of further studies.
I congratulate the authors on their work.
However, in the results section, there are probably too many investigated differences between KS patients and controls regarding metabolic profile. Authors should focus their results and discussion on differences in lipocalin2.
Author Response
Thank you for the opportunity to read and review this interesting article titled "Serum Lipocalin-2 Levels as Biomarker in Pre- And Post-Pubertal Klinefelter Syndrome Patients: A Pilot Study" by Paparella et. al.
Topic of interest in the andrological field.
The article is well written with clearly exposed results, limitations, and future and potential future directions of further studies.
I congratulate the authors on their work.
Reply: We do thank the reviewer for appreciating our contribution.
However, in the results section, there are probably too many investigated differences between KS patients and controls regarding metabolic profile. Authors should focus their results and discussion on differences in lipocalin2.
Reply: we thank the referee for this comment. Since the study was based exclusively on the measurement of circulating levels of LCN-2, it was necessary to correlate our results with the blood profile of the enrolled patients, in particular with sexual hormones, lipid and glucose profiles (known to be altered in KS patients). This comparison allowed us to discover modifications in the correlations between LCN-2 and some of the parameters, thus suggesting an involvement of the LCN-2 system in preserving the body's homeostasis. We have added, in the discussion, a paragraph that highlights the importance, in our study, of the increase in serum LCN-2 levels as a function of other metabolic parameters (page 7 lines 193-205 of the revised version).
Reviewer 2 Report
Comments and Suggestions for Authors
Dr. Paparella and colleagues explored the possibility of using Lipocalin-2 (LCN-2) as biomarker in pre- and post-pubertal KS boys. This clinical study demonstrated that pre-pubertal KS boys have increased LCN-2 levels in blood as compared to age-matched controls that are preceded the detectable changes in serum testosterone, FSH, LH and inhibin B (??) levels (this point perhaps needs to be added in the paper).
Comments and Questions:
1. It seems that FSH increased in post-pubertal boys as compared to age-matched controls in Fig 2. Is it significant?
2. Data show in Fig 3, KS boys have increased serum HDL, and decreased triglycerides levels. How this plays out in “increased cardiovascular risk” in KS individuals?
3. KS individuals have increased risk of developing autoimmune diseases. Authors perhaps need to add some discussions on this aspect since LCN-2 is involved in innate immunity.
4. What are the intra- and inter-assay CV (Co-efficiency of Variation) of LCN-2 ELISA assay?
Author Response
Dr. Paparella and colleagues explored the possibility of using Lipocalin-2 (LCN-2) as biomarker in pre- and post-pubertal KS boys. This clinical study demonstrated that pre-pubertal KS boys have increased LCN-2 levels in blood as compared to age-matched controls that are preceded the detectable changes in serum testosterone, FSH, LH and inhibin B (??) levels (this point perhaps needs to be added in the paper).
Reply: as suggested, further info was added in the revised discussion of the paper (page 6, lines 189-192 of the revised version).
It seems that FSH increased in post-pubertal boys as compared to age-matched controls in Fig 2. Is it significant?
Reply: we thank the reviewer for the observation. We added in the results section the significant difference in FSH levels between post-pubertal KS children vs age-matched healthy controls, and in the Figure 2 the post-hoc comparison (page 4 of the revised version). We remodulated the sentence in the results (page 3, lines 105-107 of the revised version)
Data show in Fig 3, KS boys have increased serum HDL, and decreased triglycerides levels. How this plays out in “increased cardiovascular risk” in KS individuals?
Reply: As we pointed out in the discussion, patients enrolled in the study surprisingly showed “reduced LDL and high HDL levels vs those of healthy controls, describing a favorable metabolic picture”. On the other hand, we found that triglyceride levels in KS individuals were below the lower limits of normality, implying a deregulation in the lipid metabolism pathway. The “increased cardiovascular risk” in KS patients affected by metabolic syndrome is observed in a variable percentage of patients as demonstrated in several studies mainly in adults (http://dx.doi.org/10.2337/dc06-0145 PMID: 16801584; http://dx.doi.org/10.1016/j.urology.2008.01.051; http://dx.doi.org/10.1016/j.fertnstert.2012.07.1122; http://dx.doi.org/10.1016/j.ijcard.2012.09.215) and few others in children (http://dx.doi.org/10.1111/andr.12275; PMID: 27802097).
The increased probability often depends on the simultaneous presence of different risk factors, In particular, hypogonadism and obesity are strictly linked to the onset of cardiovascular complications associated with metabolic syndrome (10.2174/1381612826666201102105408; 10.2337/dc06-0145).
In our cohort of patients, testosterone levels as well as BMI are in the normal range values and in line with a favorable overall metabolic picture.Nonetheless, we cannot exclude the possibility that the hormonal picture (lipid or sex-related) may change in adulthood and leave room for changes in the risk of developing metabolic syndrome. For this reason, we believe, as specified in the conclusions, it will be interesting to extend the study of LCN-2 also in a cohort of adult patients with alterations in the hormonal structure to evaluate the actual correlation with cardiovascular risks.
We added updated sentences to the discussion and new references (page 8, lines 288-297 of the revised version).
KS individuals have increased risk of developing autoimmune diseases. Authors perhaps need to add some discussions on this aspect since LCN-2 is involved in innate immunity.
Reply: as suggested, we added in the discussion a paragraph on this topic with new references (page 9, lines 298-315).
What are the intra- and inter-assay CV (Co-efficiency of Variation) of LCN-2 ELISA assay?
Reply: The supplier doesn’t perform intra and intra- assay variation tests on our DuoSet ELISA kits as they are development kits and are prone variation from user-to-user. In other words, because several key components needed to run the assay are purchased separately (such as plates, substrate, reagent diluent, etc), which could greatly impact the assay sensitivity thereby making it difficult to achieve reproducible CV values.
Based on our data, we reached inter- and intra- CV (%) ranging from 1.34% to 6.32%. Generally, inter-assay % CVs of less than 15 and intra-assay % CVs less than 10 are considerate acceptable.
Reviewer 3 Report
Comments and Suggestions for Authors
The authors of the article are thanked for the quality of the work and the choice of the subject relating to the presence of Lipocalin-2 in the serum of a group of pre-pubertal and post-pubertal children affected by Klinefelter Syndrome. Klinefelter syndrome is a male genetic disease caused by the presence of an extra X 23 chromosome, causing endocrine disorders mainly responsible in adulthood for high rate of 24 infertility and metabolic disorders. Lipocalin-2 is a small protein initially identified within neutrophils and serum Lipocalin-2 estimation seems to be a useful tool in predicting the metabolic complications caused by several pathological conditions.
According to Paparella and collaborators there is a elevated levels of Lipocalin-2 in the serum of Klinefelter Syndrome patients, compared to controls. This increase is accompanied, in pre-pubertal Klinefelter Syndrome, by the loss of correlation with LH and HDL, present instead in healthy individuals. Moreover, in all Klinefelter Syndrome individuals, a positive correlation between Lipocalin-2 and inhibin B serum concentration is found.
My only one concern is the limited size of the sample analyzed, but Paparella and collaborators realize it and, as they assure, it will be a subject of their further research.
The manuscript entitled “ Serum Lipocalin-2 Levels as Biomarker in Pre- And Post-Pubertal Klinefelter Syndrome Patients: A Pilot Study” is a good study, scientifically valid, well executed, and deserve some space in the journal. After reading the manuscript thoroughly, I have no comments to the authors. I believe the manuscript is very good and can be published in present form.
Author Response
The authors of the article are thanked for the quality of the work and the choice of the subject relating to the presence of Lipocalin-2 in the serum of a group of pre-pubertal and post-pubertal children affected by Klinefelter Syndrome. Klinefelter syndrome is a male genetic disease caused by the presence of an extra X 23 chromosome, causing endocrine disorders mainly responsible in adulthood for high rate of 24 infertility and metabolic disorders. Lipocalin-2 is a small protein initially identified within neutrophils and serum Lipocalin-2 estimation seems to be a useful tool in predicting the metabolic complications caused by several pathological conditions.
According to Paparella and collaborators there is a elevated levels of Lipocalin-2 in the serum of Klinefelter Syndrome patients, compared to controls. This increase is accompanied, in pre-pubertal Klinefelter Syndrome, by the loss of correlation with LH and HDL, present instead in healthy individuals. Moreover, in all Klinefelter Syndrome individuals, a positive correlation between Lipocalin-2 and inhibin B serum concentration is found.
My only one concern is the limited size of the sample analyzed, but Paparella and collaborators realize it and, as they assure, it will be a subject of their further research.
Reply: We are absolutely aware that the small sample size limits the significance of our results. On the other hand, we believe that the differences in serum LCN-2 levels observed in Klinefelter patients deserve to be further investigated, in the context of the study of markers of complications due to pathology. For this reason, in addition to increasing the number of pre-pubertal and adolescent patients, we will also extend the investigation to the adult population characterized by a more defined hormonal and metabolic picture.
The manuscript entitled “ Serum Lipocalin-2 Levels as Biomarker in Pre- And Post-Pubertal Klinefelter Syndrome Patients: A Pilot Study” is a good study, scientifically valid, well executed, and deserve some space in the journal. After reading the manuscript thoroughly, I have no comments to the authors. I believe the manuscript is very good and can be published in present form.
Reply: We do thank the reviewer for appreciating our contribution.
Round 2
Reviewer 2 Report
Comments and Suggestions for Authors
N/A